# The Neutrophil to Lymphocyte Ratio in Children with Bronchial Asthma

**DOI:** 10.3390/jcm12216869

**Published:** 2023-10-31

**Authors:** Ewelina Wawryk-Gawda, Monika Żybowska, Klaudia Ostrowicz

**Affiliations:** 1Department of Paediatric Pulmonology and Rheumatology, Medical University of Lublin, 20-093 Lublin, Poland; ewelina.wawryk@wp.pl; 2Students’ Scientific Group of Department of Paediatric Pulmonology and Rheumatology, Medical University of Lublin, 20-093 Lublin, Poland; ostrowiczklaudia@gmail.com

**Keywords:** asthma, NLR, neutrophil–lymphocytic index, allergy, non-allergic asthma

## Abstract

The discovery of an effective airway inflammation marker which correctly identifies the condition and phenotype of asthma still constitutes a significant challenge. The determination of NLR, that is, the ratio of neutrophils to lymphocytes, would overcome this challenge. The role of the neutrophil–lymphocytic index in the diagnosis of specific types of asthma is investigated in the present study. The results of laboratory tests of 482 pediatric patients were used for the analysis. The results of 107 children without allergic disease symptoms were selected for the control group. The mean NLR in patients with asthma was 3.42 ± 4.05, and in the control group it was 1.94 ± 1.91. The difference between the NLR in allergic and non-allergic asthma was statistically significant in the allergic asthma and control groups. There was no statistically significant difference between NLR and body temperature, BMI, and gender. The value of NLR was significantly higher in the blood of patients suffering from asthma compared to the control group. The NLR was the highest among patients with allergic asthma. The use of this blood test in daily practice may facilitate the diagnosis of asthma and differentiation between asthma types, especially when the results of other tests are inconclusive.

## 1. Introduction

Asthma is one of the most common chronic diseases among children, and in recent years there has been a sharp increase in its prevalence. It is estimated that approximately 300 million people worldwide suffer from asthma, and by 2025 this number may increase by another 100 million [1,2]. In Poland, the highest number of patients was observed in the age group of 6–10 years old [3]. The peak incidence occurs in children up to 5. In 30% of children with asthma, symptoms of the disease have already appeared in the first year of life, in 50%, symptoms have appeared before the second year of life and in 80%, sympotoms have appeared before the age of 5; therefore, both an early diagnosis and proper treatment are of great importance [4]. This disease constitutes a serious social and economic problem, since asthma exacerbations and hospitalizations are associated with high costs. The mean annual total cost per patient suffering from severe asthma is up to EUR 6500 [5].

Asthma is characterized by bronchial hyperreactivity and airway obstruction. It is defined by a number of respiratory symptoms, such as wheezing, shortness of breath, chest tightness and cough, and may have a sudden onset and a varying severity, depending on the condition of the respiratory tract, asthma control, medication intake, and exposure to allergens or infection [6]. Clinical symptoms result primarily from the inflammatory process in the medium- and small-caliber bronchi, and the consequence is the limitation of air flow into the bronchi caused by the contraction of the smooth muscle, swelling of the mucous membrane, and excessive accumulation of thick secretions, and consequently their remodeling.

The immunohistopathological picture of the condition includes infiltrates of inflammatory cells such as: neutrophils, eosinophils, lymphocytes, active mast cells and damaged epithelial cells. In people suffering from asthma, apart from inflammation of the airways, there is also a systemic activation of immune mechanisms [7]. Circulating pro-inflammatory cytokines released during the acute phase of the allergic response (such as IL-4, IL-5, GM-CSF and TNF-alpha) increase the expression of adhesion molecules, including ICAM-1 and VCAM-1 [8]. An important role in the pathogenesis of asthma (especially allergic) is played by eosinophils and Th-2 lymphocytes, although numerous studies have shown that the number of neutrophils present, e.g., in the sputum of patients, correlates with the severity and frequency of asthma exacerbations [9]. Recent research shows that many allergic stimuli can release substances from neutrophil granules in the mechanism of the formation of neutrophil extracellular traps (NETs). As a result of a prolonged NET process, tissue damage may occur, which in the case of asthma results mainly in limited airway patency [10].

An important asthma diagnostic problem, especially in pediatric patients, lies in the lack of identification of an airway inflammation marker. This would correctly identify the disease and its phenotype and significantly facilitate further prognosis. The guidelines of The American Thoracic Society, The European Respiratory Society (ATS/ERS) and The Global Initiative for Asthma (GINA) recommend that lung function tests, such as, and primarily, spirometry, should be performed to confirm the diagnosis. Unfortunately, the reliability and accuracy of this test significantly decreases in children under the age of 6, who may have difficulty following the necessary instructions [11]. Alternative lung function tests for preschool children, such as pulse oscillometry or provocative tests, are both performed only in a few pediatric centers and are characterized by a sensitivity that is too low [1].

The solution may be to determine the neutrophil–lymphocytic index (NLR), which could be a useful marker and could support existing methods. This indicator is the quotient of the number of neutrophils and lymphocytes. It has already been used in many branches of medicine, e.g., in the diagnosis of cardiovascular diseases or in the assessment of postoperative complications, in which it is an important predictor of worse prognosis, shorter survival time and the occurrence of life-threatening arrhythmias [12,13]. It is worth emphasizing that the correct value of this indicator has not yet been determined. These values vary depending on the disease classification. The NLR marker test can be made available immediately and at a low cost, and its calculation requires only basic laboratory tests.

## 2. Materials and Methods

The study analyzed the results of 482 pediatric patients admitted to the Department of Paediatric Pulmonology and Rheumatology of the University Children’s Hospital in Lublin. We conducted a retrospective study of the blood results of the patients admitted to hospital between January 2017 and December 2020. The values of laboratory parameters were obtained from the blood count taken on the first day after the patient’s admission to hospital ward. The research group consisted of 375 children. Based on the International Classification of Diseases ICD-10, the asthma patients were divided into three categories: allergic patients (J45.0, 151), non-allergic (J45.1, 151) and unspecified (J45.9, 73). The doctors classified the subjects as allergic or non-allergic according to positive or negative skin prick tests or serum-specific IgE. The control group consisted of 107 hospitalized patients diagnosed with other conditions, without a history of asthma, allergic diseases or chronic diseases. The data analysis was carried out using the Statistica 13.3 StaSoft Software. We used Student’s *t* test to analyze parametric values and Kruskal–Wallis Test to analyze non-parametric data. A receiver operating characteristic (ROC) analysis was performed. We use the area under the ROC curve (AUC) ≥ 0.5 to confirm predictive capacity.

## 3. Results

The demographic data are presented in Table 1. There was no statistically significant difference between NLR and body temperature, age, BMI, and gender (*p* > 0.05). The comparison of the mean value of the NLR coefficient between the entire study group and the control group showed that this indicator reached the value of 3.42 in the study group, which is almost twice as high as the control group (1.94). The NLR in the control group reached a clearly lower value than in each of the other examined groups, and the values of the indicator in the examined groups differed from each other. In the group of patients diagnosed with allergic asthma, the index was more than twice as high as that of the control group (4.09 vs. 1.94). It was almost twice as high (3.48) in patients with unspecified asthma (J45.9), and among children with non-allergic asthma (J45.1) the index was 50% higher than in the control group (2.72 vs. 1.07). The difference between the index in the study group and the control group was statistically significant. This difference was also statistically significant between NLR in allergic and non-allergic asthma (Table 2).

The ROC analysis indicated that proposed cut off differentiating asthmatic and non-asthmatic patients was 0.53 (according to Younden’s method), with a sensitivity of 93% and specificity of 24% (Figure 1). The proposed cut off between allergic and non-allergic asthma was 2.30 with a sensitivity of 51% and specificity of 68% (Figure 2).

## 4. Discussion

In relation to the results obtained in other publications, the present study also indicated that the value of the NLR index is significantly higher in patients suffering from asthma compared to the control group, which may indicate the usefulness of the marker in the diagnosis of bronchial asthma. This indicator was almost twice as high in the study group compared to the control group. Moreover the results of this marker present different values for different types of asthma, and the highest values were found in the group of allergic asthma patients; therefore, the NLR index in this group of patients may have the greatest significance. Their results were twice as high as those of the control group and patients with unspecified and non-allergic asthma. As in the studies of other authors, the statistically significant difference between NLR and body temperature, BMI, and gender has not been shown here. A ROC analysis indicated that the NRL cut-off point should be about 0.53 to distinguish asthmatic children from others, with a sensitivity of about 93% and specificity of only 27%, and 2.30 for allergic asthma with a specificity of 68% and sensitivity of 51%. Compared to the results of studies by M. Dogru et al. [14] conducted on Turkish patients, and in contrast to the results of studies by Ruilin Pan et al. conducted on the Chinese population [15], the present study did not prove any significance of the diversity of the values of the assessed parameters. The research conducted by Bedolla-Barajas et al. [16] demonstrates that the NLR did not differ between adult asthma patients and the control group. In the study group and the control group, the average age of patients was 33 years. Patients with asthma had higher concentrations of eosinophils and basophils, but the concentration of neutrophils did not differ from that in the control group. However, the authors conclude that inflammatory cell count ratios are gaining importance as useful indicators in categorizing asthma. Furakawa et al. [17] observed that patients with an increased number of neutrophils in sputum attended the hospital emergengy department more frequently, which may suggest the involvement of neutrophils in the inflammation of the airways and sudden deterioration of the patients’ condition. Although neutrophils participate in the host’s primary defense response, they are also responsible for the destruction of the body’s own cells and disease progression. Zhu’s research [18] also showed that the NLR index increases with the CRP parameter, especially in asthma exacerbations. The study consisted of 86 asthmatic patients, including 15 children with severe asthma, 17 with moderate asthma and 54 with mild asthma. The NLR value in the group of patients with the mild form was 2.64, while in the group of patients with the severe form, this index was almost three times higher (7.20). Another Chinese group of researchers (Guang Shi et al.) [19] found that the NLR was significantly higher in patients with critically exacerbated disease compared to patients in a stable condition and in the control group. Therefore, close monitoring for this parameter may aid in the control of the course of the diagnosed disease and the assessment of the body’s response to treatment. In the literature, it is so far recognized that high NLR values indicate a more severe course of the disease and have a bad prognosis. Ha et al. [20], in a study of idiopathic hearing loss in children, noted that NLR correlates with no recovery or moderate recovery. Guler Eraslan Doganay suggests that NLR is related to a restricted airflow in general. Its increased values were observed in patients with chronic obstructive pulmonary disease (COPD), and were associated with exacerbation, hospitalization and increased mortality [21]. It was found that the NLR could be a prognostic indicator in patients with COPD admitted to the intensive care unit.

Ruilin Pan et al., in their study at Wuxi Children’s Hospital, tested the ability of NLR to distinguish pediatric patients with asthma exacerbations from healthy children. The study group, consisting of 89 patients, showed significantly higher values of white blood cells, neutrophils, C-reactive protein and NLR compared to the control group. Additionally, parameters such as NAR (NLR–alanine aminotransferase ratio) and NBR (NLR–albumin ratio) were researched in the study. The use of a combination of these three parameters significantly influenced the selection of children with asthma exacerbation from healthy patients [15]. This study further supports the usefulness of the NLR marker assay, highlighting its numerous advantages, such as rapid testing time and non-invasiveness.

The neutrophil-to-lymphocyte ratio can be readily and easily calculated using basic laboratory tests, eliminating the need for additional visits to the doctor or extended hospital stays for the youngest patients. The conducted research provides an opportunity to expand the current diagnostic tools for asthma by incorporating the NLR index. This would enable the accurate differentiation of the type of condition in pediatric patients, thereby allowing the appropriate therapy to be initiated at the very onset of the disease. In the case of the suspicion of asthma in young children, a significant difficulty is limited diagnostic possibilities, especially the performance of additional tests, including spirometry with a bronchial reversibility test and non-specific bronchial provocation tests (exercise tests, histamine tests or methacholine). Spirometry testing is recommended from the age of 6 due to the need for the patient’s cooperation. The test is not possible in infants, and is not always possible in younger children. In most cases, the diagnosis is based only on the clinical picture and the use of trial pharmacotherapy with glucocorticosteroids, which in some cases could be avoided and is a cause of parental fear and apprehension. Therefore other markers of asthma available in GP practice are needed in this group of patients. A blood test (a complete blood count test) is one of the most popular tests, and it may be performed in a GP practice and a hospital ward. It is the basis for the diagnosis of many diseases, and is important in patients with recurrent respiratory infections. Checking the NLR may help doctors decide the treatment for the subsequent episode of wheezing. This especially applies to the youngest group of patients, whose symptoms are not very specific and therefore often underestimated. Before asthma is diagnosed, patients often are ineffectively treated with antibiotics for bronchitis, pneumonia, colds or viral diseases common in the pediatric population. During this time, the disease develops, which brings more consequences.

There are several potential limitations to this study. Other inflammatory parameters, such as CRP or ESR, were not specified. Our study was conducted retrospectively, which is why we were unable to conduct further additional tests. Secondly, patients were not divided based on the severity of their asthma. However, the ROC analysis showed quite a good specificity and sensitivity, PPV was about 0.62 and NPV was about 0.58, so we strongly believe that despite these limitations, our study may prove useful in the diagnostic assessment of asthma in the pediatric population, which could be a reason for future studies on this.

## 5. Conclusions

The presented results reveal the prospect of including the NLR marker in everyday medical practice, which would facilitate the correct diagnosis of asthma and its phenotype, as well as the implementation of appropriately selected therapy at an early stage of the disease, especially in the presence of the ambiguous results of other tests or the inability to perform these these tests on children. The ease of performance and low cost of the test may also ensure the widespread use of this method in many medical units in the future.

It is worth pointing that the calculation of this indicator may contribute to the patient’s assignment to group with a high risk of asthma development. The objective diagnosis in this age group is difficult due to the child’s lack of cooperation, and simple blood tests may allow for a quicker diagnosis. The tests, which are minimally invasive to perform, help reduce the stress and anxiety of parents, children and medical staff.

Due to the paucity of publications on the correlation between NLR and asthma, further research is needed to assess airway inflammation, systemic inflammation and the correlation between spirometry results and NLR in patients with asthma, especially in the pediatric population. Introducing this index into everyday practice may facilitate the differentiation of asthma types and risk assessment, improve prognosis, and determine the effectiveness of different therapeutic approaches. Long-term studies may be more precise and reliable in this respect.

## Figures and Tables

**Figure 1 jcm-12-06869-f001:**
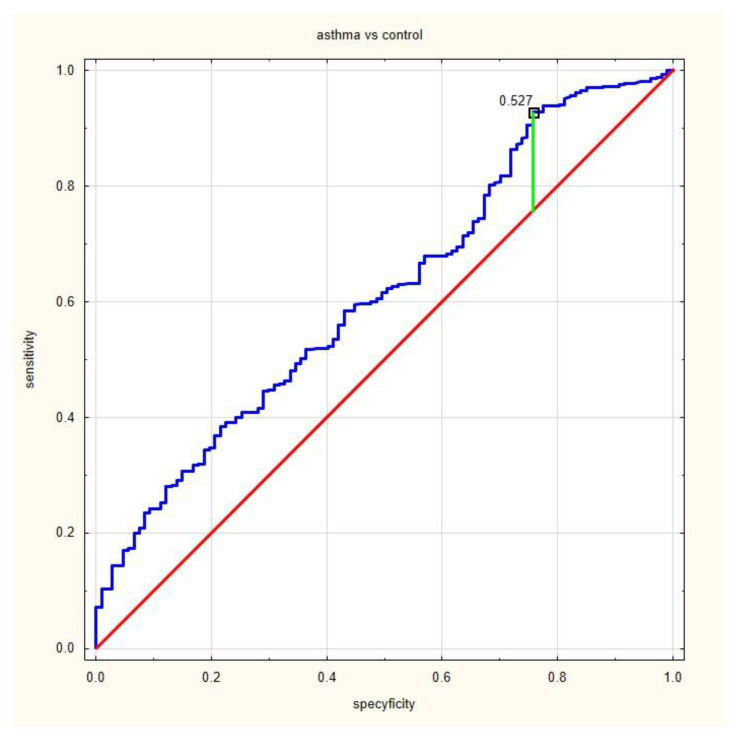
A receiver operating characteristic curve (ROC) of analysis of results of asthma patients and control; the proposed cut-off point of NRL was 0.53, area under the ROC curve (AUC) = 0.614, positive predictive value (PPV) = 0.81, negative predictive value (NPV) = 0.49, and accuracy (ACC) = 0.76.

**Figure 2 jcm-12-06869-f002:**
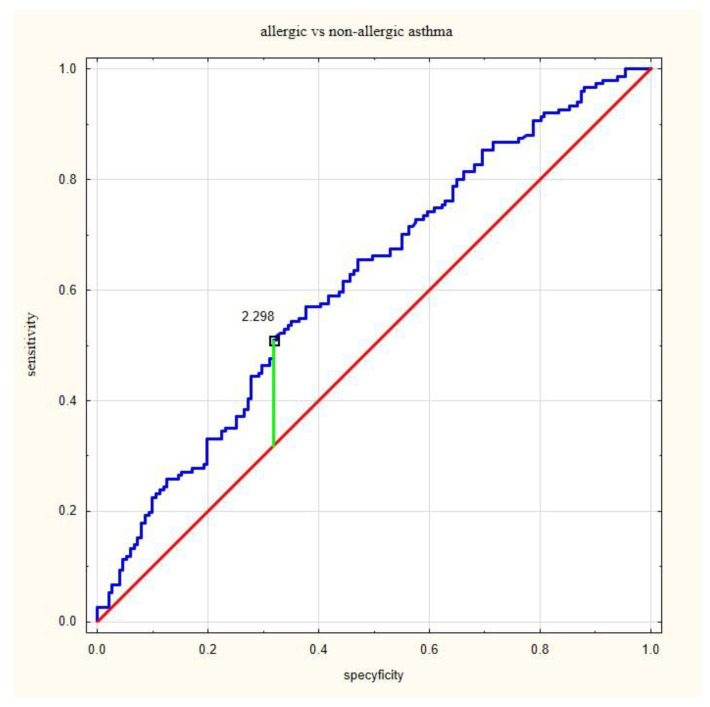
A receiver operating characteristic curve (ROC) of analysis of allergic asthma and non-allergic asthma patients; the proposed cut-off point of NRL was 2.30, area under the ROC curve (AUC) = 0.617, positive predictive value (PPV) = 0.62, negative predictive value (NPV) = 0.58, and accuracy (ACC) = 0.76.

**Table 1 jcm-12-06869-t001:** Demographics of the study population.

	Study Group (*n* = 375)	Control Group (*n* = 107)	*p*
Male/female	233/142	66/41	0.558
Age (months)			
Mean ± standard deviation	73.2 ± 49.99	44.62 ± 31.99	0.0501
Minimum–maximum (median)	5–211 (59)	2–213 (39)	

**Table 2 jcm-12-06869-t002:** Statistical analysis of the NRL index depending on the type of asthma.

Groups	*p*-Value in Kruskal–Wallis Test: H (3.N = 482) = 24.91			
	Allergic asthma	Non-allergic asthma	Other and unspecified asthma	control
Allergic asthma	x	0.00335	0.912789	0.000015
Non-allergic asthma	0.00335	x	1	0.706652
Other and unspecified asthma	0.912789	1	x	0.060373
Control	0.000015	0.706652	0.060373	x

## Data Availability

Not applicable.

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
