# Peer review of "The Neutrophil to Lymphocyte Ratio in Children with Bronchial Asthma"

_jcm, 2023, doi:10.3390/jcm12216869_

Round 1

Reviewer 1 Report

Dear Authors,

the article presents an interesting topic. There are some limitations that need to be addressed. This review provides considerations about aspects that can be improved to strengthen the study. In this study the role of the  neutrophil-lymphocytic index in the diagnosis of specific types of asthma was investigated. Principal aim of the study was to define if the use of this  blood test into the daily practice could facilitate the diagnosis and differentiation of the asthma type,  especially when the results of other tests are inconclusive.

 The structure of the article is correct but some elements should be revised and better discussed, especially the discussion, which appears brief and not very fluent. It would be preferable to discuss selection of population and possible bias, limitations and the usefulness, in clinical reality, of using this method rather than others. The article presents a statistical analysis to detect differences between NLR in allergic and non-allergic asthma and in allergic asthma and controls. However, did the author evaluate other demographic and clinical characteristics of participants? Data collection procedures are convoluted. What was the total leucocyte value? And eosinophils in individual subgroups? The blood count taken on the first day after the patient's admission to hospital ward, this could be a problem because you may have evaluated subjects in exacerbation. It could interesting to specify what is meant by unspecified asthma. Table 1 and 2 must be improved to a better readability.
Why there is not a Institutional Review Board Statement?

Hope this review will be helpful in improving the article.

Author Response

We would like to thank you for your valuable suggestions. We appreciate the feedback and opportunity to revise our manuscript entitled ‘’The neutrophil to lymphocyte ratio in children with bronchial asthma’’

Firstly, There are several potential limitations to this study. Other inflammatory parameters, such as CRP or ESR, were not analyzed in our study. The study was conducted retrospectively, which is why we were unable to conduct further additional tests. Secondly, patients were not divided based on the severity of asthma. We did not investigate the value of eosinophilia and the blood count taken on the first day after the patient's admission to hospital ward. During  hospitalization children received glicocorticosteroid, sometimes systematically, so in the next days number of leukocytes couldn’t be interpreted. We strongly believe that despite these limitations, our study may prove useful in the diagnostic assessment of asthma in the pediatric population.,

For the retrospective research without interference in hospitalization we have the approval of the hospital director and head of the  Department of Paediatric Pulmonology and Rheumatology, Medical University of Lublin.

Unspecified asthma is diagnosed in case of patient who has not yet had allergy tests performed or is suspected of having an allergy - he has symptoms, but the Prick Skin Test or sIgE test could not confirm it. 

There was no statistically significant difference between NLR and body temperature, age, BMI, and gender (p>0.05).

I look forward to hearing from you soon regarding our revised manuscript. Thank you for your consideration of this revised manuscript.  

Sincerely 

Authors 

Reviewer 2 Report

The authors have evaluated the role of NLR in pediatric asthma and in differentiating allergic asthma versus non-allergic asthma

The study is useful but there are several concerns.

Materials and methods

1. It would be useful to mention how the institute classified the subjects as allergic or non-allergic. Were skin prick tests done or serum specific IgE?

2. Line 90: other conditions need to be speicified. Are they known to influence NLR in any way?

Results: 

Table 1: shows a large difference in age between cases and controls. Is this age difference going to affect NLR results?

Line 109: use only 2 decimals (24.91)

Discussion

Line 131-132: There is no data presented that supports this statement

Stastistics: It is very important in such studies to perform ROC analysis and identify the sensitivity, specificity, PPV, NPV and accuracy

a. pediatric asthma versus controls

b. allergic asthma versus non-allergic asthma

Minor changes only

Round 2

Reviewer 1 Report

Dear Authors,

thanks for improving the manuscript. Howverer some concerns remained.It would be preferable to better explain that it is a retrospective research in materials and methods. A ROC analysis was performed. It needs to be presented and discussed in material method, result and discussion.Due to limitations of this study and lack of inflammatory markers , it would be preferable to better discuss how for a clinician,   use of this  blood test  may facilitate the diagnosis and differentiation of the asthma type.

Kind regards

Author Response

We would like to thank you for your further valuable suggestions. We appreciate the feedback and opportunity to revise our manuscript entitled ‘’The neutrophil to lymphocyte ratio in children with bronchial asthma’’.

We added information to Materials and Methods that the research was conducted retrospectively. Furthermore, we presented and discussed the ROC analysis. We believe our revised manuscript addresses your comments and is overall a stronger paper.

I look forward to hearing from you soon regarding our revised manuscript. Thank you for your consideration of this revised manuscript. 

Sincerely

Authors

Reviewer 2 Report

The authors have performed the analysis but presented only one ROC. please  present in the main manuscript ROC on allergic asthma and controls as well as non-allergic asthma and controls. All the information on sensitivity, specificity , PPV, NPV and accuracy, youdens index, AUC for all three ROC analysis should be presented in the manuscript and discussed

Author Response

We would like to thank you for your further valuable suggestions We appreciate the feedback and opportunity to revise our manuscript entitled ‘’The neutrophil to lymphocyte ratio in children with bronchial asthma’’.
In accordance with the indications presented in the review, we discussed ROC analysis. We inserted the remaining ROC curve. We discussed more of the information we were asked for.
We believe our revised manuscript addresses your comments and is overall a stronger paper.
I look forward to hearing from you soon regarding our revised manuscript. Thank you
for your consideration of this revised manuscript.

Sincerely

Authors